# Transition, Emulation and Dispute over Authority in the Bábí/Bahá'í Faith

Siarhei A. Anoshka 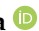

Department of Study of Religion, Faculty of Religions, University of Religions and Denominations, Qom 37185-178, Iran; anoshka@email.ua

**Abstract:** This text attempts to analyze the competition for the leadership role in the young Bábí religious community after the execution of their leader, Báb (1819–1850). With the elimination of many leaders, a small group stood out who were willing to replace the absent leader. Two preferences arose within the Babi community: forceful and pacifist. Motivated by the hunger to settle scores, supporters of the first option wanted to fight and reach the victory predicted in the Shi'ite tradition. The second option's followers, however, rejected all acts of violence, preferring to look at the Báb's texts, calling their worshipers to lofty ideals as a method of luring other people to the new religion. Presently, after the sentencing to punishment of the Prophet Báb, several people emerged among the former Shi'ites' group who made claims to authority in the community. Nevertheless, quite quickly, the main confrontation came down to a conflict between two outstanding personalities. Mírzá Yaḥyá Núrí (1831–1912), representing the radical trend of Babism, nicknamed Ṣubḥ-i Azal, was fighting for leadership with Mírzá Ḥusajn-'Alí Núrí (1817–1892), his half-brother, belonging to the peaceful Bábí party. This article describing the rivalry between two relatives for the leadership position also allows us to see the process of writing down, codifying and spreading the young Bayán religion.

**Keywords:** Báb; Bahá'í; Bahá'u'lláh; Bayán; new religious movement; religious charisma; rivalry; Ṣubḥ-i Azal





## 1. Introduction: Explanations, Transcription and Context

### 1.1. Definition of Terms

This paper includes definitions that are key terms in the discussed issue but may seem imprecise and incomprehensible to those who are not knowledgeable about the fundamentals of the Bábí/Bahá'í Faith. Thus, I consider it necessary to clarify some of the above-mentioned notions of this matter.

*al-Qá'im* (lit. 'He who will arise')—in Islam, one of the many different titles of the Hidden Imam, who is expected to return; in Shiʿite tradition, it is the visible return of the twelfth imam (Sachedina 1981). Among other titles, there are also Sáhibu'z-Zamán ('the Lord of the Age'), Ṣáḥíb al-Amr ('the Lord of Command'), al-Imám al-Muntaẓar ('the Awaited Imam') and Baqíyyatu'lláh ('Remnant of God'). It should be emphasized, however, that prophet Báb avoided openly declaring that he was *al-Qá'im*; his declarations were not very clear even for his followers (MacEoin 2009, p. 368; Naghdy 2012, pp. 37–38).

*Bayán*, or *Persian Bayán* (lit. 'expression, exposition')—the Mother Book of Bábism, written in late 1847 or early 1848 while the Báb was in the prison of Mákú (or Máh-Kú) in the mountains of Azerbaijan. This manuscript encloses elements of religious law and a discussion of doctrinal concepts and anticipates the World Order of Bahá'u'lláh. It was one of the Báb's first opera in which he clearly states that he is the messianic figure of the Hidden Imam (*Mahdí*) that the Shi'ites were waiting for (Amanat 2007, pp. 337–50). In these principal scriptural writings, the Prophet Báb declares himself the supreme Mirror of God and explains that his disciples are worldly Mirrors (Saiedi 2008, pp. 270–71). In this doctrinal work, Báb introduces an unambiguous division between the *People of the Qur'án*

and the *People of the Bayán*. The latter have the religion of justice, Báb asserted (Иоаннесян 2011, pp. 208–9; Khan [1986] 2007, pp. 247, 267).

## 1.2. Transcription Issues

The text below contains quite a number of terms and words of Persian origin, which have been transcribed in accordance with the unified format adopted by the author in his other texts on the matter. However, it should be recognized that other renditions of transcriptions can also be found in the literature on the subject (Table 1). The table below contains equivalents to the most common and important terms occurring in the text of the article according to the transcription of Encyclopædia Iranica.

**Table 1.** Index of transcriptions of some key phrases of Persian and Arabic origin.

| Encyclopædia *Iranica* | Article Text | Encyclopædia *Iranica* | Article Text |
|---|---|---|---|
| Mīrzā Ḥosayn-ʿAlī Nūrī | Mírzá Ḥusajn-ʾAlí Núrí | ʿAbd-al-Bahāʾ | ʾAbdu'l-Bahá |
| Mīrzā Yaḥyā Nūrī | Mírzá Yaḥyá Núrí | Bāb | Báb |
| Moḥammad | Muḥammad | Babi/Bahai Faith | Bábí/Bahá'í Faith |
| Shah Nāṣer-al-Dīn | Shah Náṣiri'd-Dín | Bahāʾ-Allāh | Bahá'u'lláh |
| Shaikh Aḥmad Aḥsāʾī | Shaykh al-Aḥsáʾí | Bayān | Bayán |
| Shiʿite | Shiʿite | Ketāb-e aqdas | Kitáb-i-Aqdas |
| Ṣobḥ-e Azal | Ṣubḥ-i Azal | Mehdī | Mahdí |
| Qajar | Qájár | man yoẓheroh Allāh | man yuẓhiruhú Alláh |
| ʿolamāʾ | ʾulamʾ | maẓhar-e elāhī | maẓhar-i Illáhí |

## 1.3. Religious and Social Context in 18th–19th Century Persia

At the end of the 18th century, major changes took place in Shiʿite reflection. A few decades later, it would become the base of the Bábí movement. For the previous two hundred years, the ulemas had tried to make Shiʿism the nationwide religion of Persia by strengthening their branch of Islam (known as *Twelver*). However, the Ismaʿili monarchs (otherwise known as *Sevener*), who traced their lineage back to the seventh imam, constantly justified their privilege to rule. Especially threatening steps for the Twelver were the shah's attempts to deprive the imams of their role as interpreters of the Qurʾánic Law—a legislative tool of *ijtihád* (Adams 1994, p. 351).

There has always been acute skepticism about any form of secular governance in Shiʿa theology. Thus, as the ulemas declared, they had the authority as spokespeople of the believers' community. Clergy seemed to be the embodiment of charisma with a double dimension: hereditary and acquired. The clergy's claims to authority were motivated by the fact that religious elites maintained continuity despite political turmoil. To better understand the origins of the emergence of the Bábí movement in 19th-century Persia, it is required to draft two different lines of the Shiʿite religious class.

The disputed issue between the intellectual traditions was a religious authority. In which way, by whom, and on what principles should matters relating to the faith and practice of Islam be resolved? The disagreement is named after the names of both groups and is known as the dispute between the Akhbarites (traditionalists) and the Usulites (rationalists). The Akhbarites maintained that contentious issues were to be resolved according to the teachings and practices of the Prophet or the imams. The Achbarites regarded scripture and tradition to be the basic sources of knowledge. The Usulites, their group of opponents, argued that Islamic law contained particular principles which, once internalized, could then be applied to comment on tradition (al-Bakhati 2021; Fazlhashemi 2010).

In the context of our reflections, it is important that another source of law for the Usulites was the unanimous opinion of the ulemas, as well as the principle of analogical reasoning. Therefore, the Shiʿi jurist (known as *mujtahid*) was given special respect, and followers were ordered to submissively obey him in matters of religious law (Bayat 1982, p. 21). Such a perception of the mujtahids made them the exclusive custodians of morality and the truths of faith. Although all members of the clergy had splendor, only a few of

them were recognized as mujtahids *sui generis*, distinguished by the group of disciples, and retained authority and influence (Adams 1994, pp. 353–54).

Over time, the tendency grew among the Usulites to perceive one among the mujtahids as superior to the others in the interpretation of the law and piety. In the mid-19th century, expectations for a new Messiah amplified: 1844 was the 1000th anniversary of the Hidden Imam's disappearance. There was a widespread belief that humankind was to learn about the Savior's advent from a precursor—a perfect Shiʿite, who would be called the Báb (lit. 'the Gate'). Such a someone was Sayyid ʿAlí Muḥammad (1819–1850), an autodidact who from an early age was famous for composing a large number of prayers, speeches, and theological treatises. A young merchant from Shíráz proclaimed himself the Gate of God—the Báb—and later even the Hidden Imam (Momen 2003, p. 330).

It seems beyond dispute that such proclamations were a complete break from Muslim orthodoxy. These declarations could no longer be tolerated by both the ulemas and the civil government. The growing popularity of the pro-Báb movement led to his imprisonment. Declaring himself a prophet further fueled the conflict between the ulema elites and a new heretical movement led by a charismatic leader. The Báb's subsequent execution just reinforced the eschatological expectations among the believers, causing a rush of claims for leadership in the new community.

## 2. The Case Study: The Struggle for Leadership and the Story of the Baháʾí Leader's Charisma

The Bábí religious movement in Persia was brutally pacified, but the religious renewal idea did not disappear (Ghadimi 2009, p. 16). Undoubtedly, the loss of their leader weakened their activity, although the Báb repeatedly proclaimed that he would not finish the assignment he had initiated because he would be replaced by another person whose words were better than his "thousand Scriptures" (Лалуев 2016, p. 229). Therefore, the need to consolidate the flimsy refugee community became an urgent problem.

In the context of the reflection undertaken, our attention is drawn not so much to the problem of faith preaching but rather to the issue of succession after the tormenting and execution of Babism's founder. Since the Bábí movement in its first phase was not based on a specific doctrine but rather depended on the charisma of its founder, the Báb's demise brought frustration and despondency (Scharbrodt 2008, pp. 36–37). It should not be surprising, therefore, that with the elimination of many leaders, a small group of people stood out, wanting to replace the absent leader. Despite significant emigration from Persia, there was a vigorous community of the Báb's followers in Tehran, within which, several individuals aspired to take on the leader's role (MacEoin 2009, pp. 376–78).

Two tendencies arose within the Bábí community: forceful and peaceful (Geaves 2009, pp. 38–39; Johnson 1992, p. 12; Prophet 2016, p. 46). Encouraged by the desire to take revenge on their enemies, the first option's adherents desired to fight and achieve the victory predicted in the Shiʿite tradition. The second ones, however, rejected all acts of brutality, preferring to scrutinize the Báb's texts, calling their worshipers to sublime ideals as a method of attracting others to the new religion (Momen 2008, pp. 64–65). It was not easy to articulate a clear division, as representatives of both coalitions could obey meetings of each of these groups.

The most imperious demand for omnipotence was the mullah Sheikh ʿAlí Turshízí, called ʿAzim, or the 'Mighty'. As one of the most radical Bábís—he tried to organize an action to extricate the Báb—he depicted the militant wing. Another aspirant was the blind, eloquent orator Sayyid Baṣír Hindí, originating from Sufi background, who claimed that his writings were inspired by the Báb. Focused on Hindu beliefs, especially those about reincarnation, he proclaimed himself Imam Ḥusajn. These two opponents vied with each other until Baṣír Hindí left the capital city for the southeastern part of the country, where he was assassinated (Cole 2004, p. 228).

Mírzá Yaḥyá Núrí (1831–1912), nicknamed Ṣubḥ-i Azal, i.e., the 'Morn of Eternity', representing the radical direction of Babism, also fought for leadership in Tehran (Bromley

2016, p. 110). Ṣubḥ-i Azal was in constant contact with the Báb starting from 1848, whose papers contained numerous references to him. This was supposed to prove that Azal was perceived by him as his successor and forthcoming leader (Buck 1995, p. 72; MacEoin 2009, p. 281; Momen 2007, p. 87). It is no coincidence that Azal received honorary titles such as 'Everlasting Mirror', 'Name of Eternity', and 'Fruit of the *Bayán*'. According to the Báb, the probable reason for naming Azal as successor was the young man's mastery in receiving divine knowledge and revealing the meaning of scripture verses (Warburg 2006, p. 146). The Báb advised Azal to keep what was revealed in the *Bayán* for he was the "Great Way of Truth" (Smith 2000, p. 53).

Unlike his rivals, who made radical theophanic claims similar to those made by the Báb himself before his death, Ṣubḥ-i Azal advocated a more balanced approach to matters of charismatic authority. Both he and his adherents accentuated the conservative nature of the revelation and focused on the doctrines of the *Bayán* and other later scriptures (MacEoin 1983, p. 220).

The next succession rival was Mírzá Ḥusajn-'Alí Núrí (1817–1892), Azal's half-brother, nicknamed Bahá, meaning 'Glory', belonging to the peaceful Bábí section. Like the Báb, Ḥusajn-'Alí, despite his noble bloodlines, did not receive the proper education that would allow him to partake in philosophical or theological debates. Regardless, his biographers convey that his outstanding abilities, wit, and discernment allowed him to deliver sermons that were listened to by a large number of clergy and theologians. The ulemas, scholars of Islamic doctrine and law, were amazed, assuming it was a miracle (cf. Luke 2:47; Иоаннесян 2003, p. 55).

As the high-ranking officeholder's son, who had joined the Bábís in 1844, Ḥusajn-'Alí made no formal avowal to dominion, although in many respects he was the arranger of the Tehran group. Having lost his father at the age of 22, the prospective leader of the new religion inherited an impressive fortune, extensive land, and family estates, which allowed him to generously finance the movement's activities and support many of his companions of faith who were deprived of their livelihoods due to persecution and repression (Browne [1892] 1999, p. 349; Пивоваров 2014, p. 32). His house was a meeting point for Bábí disciples and a place to stay for pilgrims.

Due to his tenderness and compassion for the oppressed, despite his very young age, he was dubbed by the people the 'Father of the Poor' (Bahá'u'lláh 2011, p. 26, 41; Hartz 2009, p. 38; Terry 2008, p. 104). At the beginning of 1851, on the orders of the Grand Vizier, Bahá went on a "pilgrimage to Iraq"—the euphemistic term for banishment aimed at expelling someone from Persia. The expulsion of one foe who was inhibiting the more extreme and excitable elements in the Bábí community inevitably resulted in a shift of the balance towards the extreme trend. 'Azim and Azal began to plot against power to overthrow the Qájár dynasty and establish a Bábí state as part of the Báb's political claims (Грачева and Мартыненко 2015, p. 53; Лалуев 2016, pp. 225–26; Пивоваров 2014, pp. 20–22). Upon learning of this, wishing to avoid trouble, Ḥusajn-'Alí decided to call Azal to order, but the latter refused to obey.

Meanwhile, Ḥusayn Ján Mílání emerged as a young, charismatic weaver in Tehran, under whose guidance a few, although the most radical, followers gathered. This group decided to murder the Shah, which was presumed to be an apocalyptic event, and which would be accompanied by the intervention of supernatural forces with the subsequent founding of the Bábí state (Saliba 2003, pp. 144, 171). Tempers were spurred by the proclamation of Ḥusayn Ján as Imam Ḥusayn (Momen 2008, p. 65). At the same time, 'Azim and Azal were stockpiling weapons, sending appeals to their fellow believers asking them to come to Tehran to help carry out their assassination plan.

The change in the office of the Grand Vizier resulted in a temporary thaw. In May 1852, Ḥusajn-'Alí returned to Tehran. This was due to the new vizier's desire to prove himself to the monarch by controlling the situation with the Bábís. After a transient stay in the capital, Ḥusajn-'Alí was sent back to the province, so he had no communication and could not influence the decisions made in the community, which would soon bring on

the Bábís' persecution that was more fierce than those they had previously experienced (Browne [1892] 1999, p. 347).

Ḥusayn Ján's sermons became more and more exciting, stimulating the listeners' emotions to such an extent that his charisma even enticed the ruling dynasty princes to the meetings. It seemed that most adepts tolerated his claim to a higher status and were even willing to reform the laws based on the Báb's writings. Even Azal, who was one of the initiators of the collusion against the Shah, described Ḥusayn Ján as "*the most turbulent and eager for mischief and yet the most pusillanimous of those who professed to follow the Báb*" (MacEoin 2009, p. 382).

An unsuccessful attempt on the shah's life on 15 August 1852 ended with one of the attacker's deaths and the other two's arrests. The tormented would-be regicides admitted under torture that they belonged to the Bábí movement, which gave the regime an impetus to launch large-scale brutal repression. Shah Náṣiri'd-Dín ordered the arrest and execution of all the Bábís in the country, although just a small group—from thirty to seventy people— were responsible for this mutiny. The situation was made more threatening by an attempted rebellion in the north of the country by 22-year-old Ṣubḥ-i Azal (Zabihi-Moghaddam 2004, p. 181).

The Bábís condemned the attack on the monarch, regarding 'Azim as the main perpetrator, a "*madman deranged by grief over the* Báb's death" and as "the cause of shame to mankind" (Bayat 1982, p. 128). The Bábís' repentance did not achieve their goal; the month of August was marked by the continuous increase in executions. The Báb's followers were beaten, daggered and cut into pieces, flayed with whips, forced to eat their ears, blinded, shod like horses, had lit wicks placed in their wounds, and had their teeth extracted and then driven into the tops of their heads (Browne [1892] 1999, p. 348; Грачева and Мартыненко 2015, pp. 59–60).

Ḥusayn Ján and 'Azim were convicted, and several of the leaders went to jail. Ḥusajn-'Alí spent four months in a semi-dark, worm-infested, and disgusting cistern which was turned into a prison cell, called the Black Pit (Hartz 2009, p. 42; Lepard 2008, pp. 31–34). According to the prevailing Bahá'í narrative, it was here that Ḥusajn-'Alí experienced his first vision, experiencing the Spirit of God's presence in the form of a heavenly virgin (cf. Revelation 12:1a), who assured him of his divine mission and promised help (Иоаннесян 2003, pp. 61–63; Hutter 2009, p. 32; Warburg 2006, p. 147). This event—similar to Muḥammad's visitation by an angel or Moses' consternation during his vision of God—is considered by Bahá'ís to signify the birth of the revelation of the Faith, which is the center of Bahá'u'lláh's claims about his Mission.

Baghdad, for the next dozen years, became the new home of the Bábís, who escaped terrible repression. The center of the movement shifted from Persian territory to Ottoman territory. The persecution continued with varying intensity, so the Baghdad diaspora was constantly filled with exiles pushed to leave their homeland. The regime of the Sultans turned out to be more indulgent towards the Bábís than the monarchy of the Shahs, which managed to almost completely get rid of troublemakers and insurgents from its territory.

Bábís leaders living in these new locations, where intense rivalry would soon arise, became involved in writing down, codifying, and disseminating the *Bayán* religion. They engaged in encouraging followers to refrain from all resistance to the authorities, but through righteous living, patient resignation, and kind manners towards all people, to present their faith to the whole world (Browne [1892] 1999, p. 350).

Sick and exhausted, Ḥusajn-'Alí, supported by the Russian diplomatic mission, was released after proving his blamelessness, sentenced to exile, and, with the court's permission, went to Baghdad. Mírzá Yaḥyá Núrí also escaped to Iraq. He avoided arrest by hiding in the mountains and pretending to be a dervish while remaining the nominal head of the Bábí community. Although Ṣubḥ-i Azal joined his brother, he lived in seclusion, avoiding any Bábís who wanted to have contact with him. It is speculated that Mírzá Yaḥyá hid himself for security reasons, to avoid possible dangers arising from his position in the community (Buck 1995, p. 72). As he retreated, Mírzá Yaḥyá increasingly lost leverage over

the community, who saw, although he was still esteemed by them, that the group could cope without him (Hutter 2009, p. 33).

Some members of the Bábí group began to undermine Azal's authority, claiming the right to be Imam Ḥusayn and even declaring themselves the *One Whom God Will Make Manifest*, foretold by the Báb, vehemently demanding that he relinquish his insistence of supremacy. Azal's self-isolation was criticized by Bahá himself, who, however, still recognized his right to lead the community, supported him financially, and at the same time surrounded himself with a loyal bunch of disciples, doubting his brother's lead (Грачева and Мартыненко 2015, p. 62). Tensions between the brothers inevitably increased. Ḥusajn-'Alí decided to go to the Sufi monastery in Kurdistan, where he prayed and meditated, dressed as a dervish, and led an ascetic, solitary life for two years (cf. Exodus 23:15–18; Luke 4:1–2; Surah 97:1–5; Brogan 2003, pp. 8–10; Scharbrodt 2008, p. 38).

The period of 1852–1863, when Azal remained in hiding, is very little explored in Bábí/Bahá'í studies. This is explained by the insufficient number of reliable sources and their predilection depending on their authors' affiliation (Bayat 1982, pp. 127–28). A further difficulty is the not fully clarified role of Ṣubḥ-i Azal's texts, the scriptures that are often omitted when discussing the history of that decade (MacEoin 1992, pp. 38–41). There is a shared statement among Bahá'ís that Azal's pieces are childish, disjointed, and meaningless (Wilson [1915] 1970, pp. 47, 183–85).

All this does not make it any easier to analyze the imprecise aspects of the doctrine and the Faith's history. Conspiracies, collusion, and petty altercations, as well as the Bábís' activities, worried the Persian authorities. In 1863, they persuaded the Sultan to move the anarchists and nihilists—as the Báb's disciples in the Qájár realm were called—further from the Persian borders. It was then, after a series of mystical cases, that in the spring of 1863, before his family and a handful of loyal followers, Bahá declared that he was a Manifestation of God, the proclaimed continuator of the Báb's mission, whose advent was heralded by the *Bayán*. In Bahá'í terminology, this was called the lift of the veil (cf. Matthew 27:51a): the Báb revelation was officially halted, and the dispensation of the new faith began (Stockman 2013, p. 94; Грачева and Мартыненко 2015, pp. 62–63). Four years later, in exile in Edirne, Ḥusajn-'Alí would publicly repeat this declaration, taking the epithet Bahá'u'lláh (Arabic: بَهاء آلله), meaning 'The Glory of God' (Berger 1957, p. 99). From March 1866, another name for the believers' community also came into force: the Bahá people or Bahá'í (Buck 2011, p. 75; Saiedi 2000, p. 177). It seemed to be true that the Declaration of 1863 was timely because it laid the foundations for the charisma that a flimsy community needed.

Bahá'u'lláh, calling himself *Man yuẓhiruhú Alláh* (*Him whom God shall make manifest*), formally proclaimed himself the new head of the community of believers, predictably facing direct opposition from Ṣubḥ-i Azal and his backers (Chryssides 2012, pp. 198–99). It was commonly believed that a new divine manifestation would not appear until a thousand years later when the Báb's religion would be permitted in many nations. It is true that, tempted by the universal nature of Bahá's revelation, numerous followers of Zoroastrianism and Judaism living in those areas converted to the new faith (Buck 2004, pp. 147–50). Since the prevailing conditions did not meet the followers' expectations, including those from other denominations, they were able to switch and redirect attention to the charismatic novelty of Bahá'u'lláh. In this way, religious newcomers wished to identify with the new group's norms rather than with their previously defined standards.

The endeavor to defend the tenets of traditional Babism proved unpopular: over time, many Bábís came under the authority of Bahá'u'lláh. All those who decided to stick with Bahá'u'lláh's brother became a minority, called Azalites, and their doctrine was called Azalism (proper name: Bayáni faith or Azali Bábí) (Browne [1892] 1999, pp. 350–51; Landowski 2008, p. 31). Thereafter, official chroniclers of the Faith would begin the process of obliterating Azal from history, denying his popularity during the period of 1854–1865. Azal was even accused of trying to poison his brother with mercury chloride out of envy, as a result of which, the Bahá'u'lláh's hair turned white and his hand trembled for the rest of his life (Bowers 2004, pp. 61–62; Smith 2000, p. 76; Wilson [1915] 1970, pp. 224–27).

Bahá'u'lláh called the events of the 1860s the great division, and in the Bahá'í Faith, the term *náqiḍín* (lit. 'Covenant-breakers') was coined to describe apostates (cf. Romans 1:31; Cederquist 2005, p. 185). According to the hagiographic narrative, the Divine Teacher Bahá not only overcame his brother's selfish tendencies but indeed always had his superiority (Иоаннесян 2003, pp. 57–59).

Ḥusajn-'Alí, also known as Bahá'u'lláh, wrote many tablets—tractates and letters in the form of revelation. In those texts, he presented new claims to be the reappearing Báb, thereby replacing any authority of Azal as the supposed Báb's heir. Soon, Bahá'u'lláh denied that the Báb had ever actually appointed an inheritor, even though most Bábís so far perceived Azal as just this type of person (Cole 2004, pp. 231–32; Wilson [1915] 1970, pp. 204–5).

The late summer of 1867 was the culmination of a dispute between two brothers. Invoking the idea of *rajʿá* (Arabic lit. 'return'), that is known in Shiʿa doctrine, Bahá'u'lláh declared himself to be the return of the Prophet Muḥammad. By redefining the authority of prophets, messengers, or imams as supporting the Cause of God, that is, Bahá'u'lláh himself, the Bahá'í leader ultimately distanced himself from Ṣubḥ-i Azal. Moreover, since his exile in Edirne, Bahá'u'lláh began to send letters to powerful figures in the world, including Emperors Louis Napoleon III and Alexander II, Queen Victoria, and even Pope Pius IX. The ambition of this correspondence was to explain the mission and declare the dawn of a new era, as well as a call to common action for world peace (cf. Luke 2:14), and at the same time, it was a bid to Christians to recognize Christ in his person (Berger 1957, p. 100; Buck 2004, pp. 157–72; Иоаннесян 2003, pp. 88–91; McLean 2008, p. 246). Ḥusajn-'Alí addressed the pope as the Messiah who established the laws of humanity's spiritual rebirth; Bahá'ís saw this as the fulfillment of Christ's parousia promise (Лалуев 2016, pp. 262–65).

Fierce disputes between the two factions, leading to violence on both sides, the influx of refugees from Iran, and proselytism among the Muslim population goaded the Ottoman authorities into expelling the feuding Bábís in 1868. Together with his allies, Bahá was sent to the penal colony of Acre (or 'Akká) in Ottoman Syria, while Azal, his family, and several adherents were deported to Famagusta in Cyprus (Bahá'u'lláh 2011, p. 102; Wilson [1915] 1970, pp. 22, 224–33). After two years in prison, Bahá'u'lláh spent the next several years under house arrest. After the death, his body was placed in a tomb in the gardens of the Bahjí residence on the outskirts of Acre, where the founder of the new religion stayed in the last decade of his life (Adamson 2007, p. 58). Similarly to the sanctuary on Mount Carmel in Haifa (The Shrine of the Báb), Bahjí—inscribed on the UNESCO heritage list in 2008—is a point of attraction for masses of Bahá'í pilgrims all over the world (Hartz 2009, pp. 84, 110, 138).

The separation compelled upon the brothers had fatal consequences for Mírzá Yaḥyá: a lack of resources and isolating conditions on the island separated Ṣubḥ-i Azal, precluding him from communicating with other Bábí groups (Hutter 2009, p. 41). There was no longer any deep expectation of restoring congruence among the Bábís. Mírzá Yaḥyá left neither a will nor an express order as to his successor: after he died in 1912, no one equaled him in charisma or intelligence. Azali Bábí entered a period of stagnation from which he never recovered (Landowski 2008, p. 31; MacEoin 2009, pp. 594–95). The grave of the founder of the Azali schism (The Shrine of Ṣubḥ-i Azal) on the outskirts of Famagusta, Cyprus, is occasionally visited by a few followers who remain faithful to him to this day.

Impacted by Christian ideas and contacts with European missionaries, Bahá'u'lláh became increasingly charmed by Western conceptions (Abassy 2010, pp. 90–91; MacEoin 1983, p. 227). While staying in Edirne, Bahá began to create the *Kitáb-i-Aqdas* (Arabic: الكِتّاب الأقدَس; the Most Holy Book). This, the Bahá'í Faith's most important scripture, was an amalgamation of modified Báb postulations filled by the new doctrines from Bahá'u'lláh. Completed in 1873, the *Kitáb-i-Aqdas* is of essential importance because in it, Ḥusajn-'Alí enumerated seven sources of authority relevant to his religious community's followers (Cole 2005, p. 333).

The first in the hierarchy was the person of Bahá'u'lláh, the Prophet of the new religion (Bowers 2004, p. 273; Buck 1999, pp. 215–17; Schaefer 2009, pp. 535–36). The Azali group, whose leader never called himself a prophet, rejected the claims of Ḥusajn-'Alí, reasoning that the Divine Essence could not reveal either a new prophet or a new scripture until the mission of the previous prophet, i.e., the Báb, would be durable. One of the Azalite arguments was that Bahá'u'lláh abolished many of the doctrines and practices of his predecessor, even though the Báb was still viewed as the 'Herald *of the Day of Days'* (Effendi 1991, p. 123; Terry 2008, p. 226). Indeed, the Báb's subordinate laws and commandments were abolished, which Bahá'u'lláh attributed to the permission he had received from his predecessor. Bahá explained this by saying that each new prophet has to adapt his teaching to humankind and the specific circumstances of the age at the revelation moment. Referring to C. Jung's concept of archetypes, Bahá'u'lláh in a sense—figuratively speaking—clothed himself in the robes of the Wise Old Man (Sage), being not only a teacher transmitting the faith but also an originator who initiated the process of true discernment of his predecessor legacy.

The second rank on the authority scale was the writings of Bahá'u'lláh, 'every single letter proceeding from Our mouth', which was called the creative word of God. These texts are still considered a remnant of Bahá'u'lláh's charisma, being a source of inspiration and edification for his devotees (Buck 1999, p. 141; Wilson [1915] 1970, p. 115). Third in rank was 'Abbás, the eldest son, who would succeed to the headship as 'Abdu'l-Bahá after his father's death. In the longer term, 'Abdu'l-Bahá would become one of the three—with Báb and Bahá'u'lláh being the most important faith figures, his scripts and speeches would become part of the canon of Bahá'í sacred literature (Adamson 2007, p. 513). The fourth hierarchy rung was taken up by the *aghṣán* (Arabic: أغْصان; lit. branches)—the male descendants of Bahá'u'lláh. They deserved special esteem and courteousness, although they had no—neither administrative nor spiritual—majesty (Smith 2000, p. 30).

The fifth most important grade in the community was the entire people with their representatives. The sixth level was regaled by the ʿ*ulamá'*, or scholars of the Bahá'í community. Finally, the last, seventh place in the management system was occupied by the Universal House of Justice (UHJ), the nine-member supreme ruling body. These sources of authority had different spheres of responsibility and commitments, and no single source of authority was perceived as having control over all issues within religion. In this way, Bahá'u'lláh initiated the process of charisma institutionalization. This would have conditioned the functioning of the anticipated ideal society of the future. Citing psychologist Le Bon, it can be concluded that Bahá'u'lláh used a compensation strategy, entering into the optics of a future-appropriate life rather than intervening in temporal life.

In the *Kitáb-i-Aqdas*, Bahá'u'lláh explains the hierarchy of authority as follows: should differences arise amongst you over any matter, refer it to God while the Sun still shineth above the horizon of this Heaven and, when it hath set, refer ye to whatsoever hath been sent down by Him (Bahá'u'lláh 2005, p. 16). It follows that the Bahá'í leader was initially convinced that his person [the Sun] and his texts were sufficient for believers in matters of hermeneutics or interpretation of the scripture canon. Bahá'u'lláh can be viewed as M. Weber's '*bearer of the charisma*', whose mission is to preach (Stone 2016, pp. 122–23).

Contrary to the Muslim concept of spiritual and secular authority being united in the person of the Prophet and the *Mahdí*, Bahá'u'lláh's charismatic claims were bereft of any political expectations and were limited to spiritual authority (Rieff 2008, p. 54; Scharbrodt 2008, pp. 14, 60). While Muslim reformers desired a return to the classical order, the idea of the Bahá'í Golden Age proposed the construction of a future world order, comprehended in terms of civilization, which was to become the fruit of a new revelation (Dehghani 2014, p. 16; Effendi 1991, pp. 98–99). The living prophet was the spiritual center of the religion, its administrative core, with a small bureaucracy in the city of Acre, maintained by donations from the disciples (Bahá'u'lláh 2017, pp. 247–49).

For Bahá'u'lláh, the issue of succession was not so much a problem of transferring authority to heirs, but rather one of defending the teaching orthodoxy and then passing it

on to following epochs. The Bahá'í leader wrote the following words about his son: when the Mystic Dove will have winged its flight (...) and sought its far-off goal (...), refer ye whatsoever ye understand not in the Book to Him Who hath branched from this mighty Stock (Bahá'u'lláh 2005, p. 50). 'Abbás played a pivotal role, mediating between his father and the external world and becoming the unerring exponent of the scriptures' meaning. 'Abbás Effendí, also known as 'Abdu'l-Bahá, called upon the disciples to obey him as the Center of the Covenant appointed by the Lord of the Covenant [i.e., Bahá'u'lláh], the sole authoritative interpreter of his father's teachings and the *source of authority* (Hatcher and Martin 2002, p. 50; Stockman 2013, p. 122).

Turning attention to Ḥusajn-'Alí's three other sons, it appears that they were not initially intended to fulfill any detailed role. However, there was a group of Bábís who attributed particular charisma to one of Bahá'u'lláh's sons. However, in the *Kitáb-i-Aqdas*, when discussing endowments, the author mentions that after him, "the authority shall pass to the Aghṣán" (Bahá'u'lláh 2005, p. 13). The Bahá'í chairperson had in mind distinctly only sons, not further descendants, a family line that could continue ad infinitum. The superior of the Faith wanted the House of Justice or a general body of viceroys to be established at the moment of the death of his last son (Cole 2005, p. 315). The custom of ancestral charisma possessed by the Prophet's alleged descendants was hereby questioned. However, all four of Bahá'u'lláh's sons received the dignified epithets of the earlier prophets—Jesus, Muḥammad, Abraham, and Moses (Wilson [1915] 1970, p. 250).

Because the religion of Bahá placed a decisive emphasis on promoting world civilization, there are certain areas in which the authority to implement its principles rests not with the authorities of the Faith but with the people and their representatives. In proclaiming the separation of religion and state, Bahá'u'lláh renounced all theocratic claims, anticipating that "the reins of power fall into the hands of the people" (Bahá'u'lláh 2005, p. 28). Considering that specific principles and laws of a social nature could only be implemented by civil legislatures, Bahá'u'lláh relinquished authority over the peoples and their delegates in the parliaments. In Freud's terminology, it would be the process of uniting a group of many equal fellows who can identify with each other, thereby intensifying the process of experiencing their faith.

Referring to religious knowledge, the Bahá'í leader saw charisma in believers, commensurately making them carriers of divine inspiration to the extent that they could learn to answer spiritual queries, without rejecting the knowledge of the sacred texts. Whether individuals or groups, they were to serve both individual believers and the entire community, because they are channels of divine guidance (Danesh 2014, pp. 1–2; Sadeghzade Milani 2002, p. 100).

Bahá'u'lláh experienced permanent exile: forty of his seventy-four years of life were spent in exile and emigration. Bahá'u'lláh comprehended that through converts from Shiʿism in Persia, despite his absence, he could also rule in this region. A loyal corps of like-minded people then formed around him. Over time, they turned out to be invaluable preachers of the new faith among Iranians. Just as the Báb did by giving honorable religious titles to his closest allies, the Bahá'í Faith founder also bestowed theophoric epithets on selected propagators of his teachings, giving them an almost theophanic aura (Smith 2000, p. 185).

Such an approach was not merely flattery towards the converted ulemas but was based on the idea of the parallelism of the virtues of these people to the divine traits (Saiedi 2008, pp. 335–36; Schaefer 2009, p. 7). In the 1880s, as their ranks extended, the Bahá'í ulemas gained increasing esteem within the believers' congregation. Since Shiʿite Iranians were accustomed to entrusting religious leadership to learn from someone valued for eloquence and persuasion, it was natural for Bahá'ís to turn their aspirations to the ʿulamá'. This is what Ḥusajn-'Alí himself wrote about them: Happy are ye, O ye the learned ones in Bahá [...]. Ye are the billows of the Most Mighty Ocean, the stars of the firmament of Glory, the standards of triumph [...]. Well is it with him that turneth unto you, and woe betide the froward (Bahá'u'lláh 2005, pp. 49–50).



Bahá'u'lláh planned a corps of Shiʿa-style ulemas to have considerable leverage within the Bahá's community as well, because madrassa-trained converts were well equipped to interpret the Arabic scriptures to laypeople in Persia. Members of the theological class of eminently high status were regarded by Bahá'ís to be pathways to God. Towards the end of his life, in 1886, Bahá'u'lláh established four of them as the Hands of the Cause of God—a title that was both a rank and an administrative function in the defense and advancement of the Faith (Adamson 2007, pp. 217–20, 527–28; Hudson 1986, pp. 521–27; Hutter 2009, p. 54; Stockman 2013, pp. 100–1).

Even during the lifetime of the founder of the Bahá'í Faith, an institution was created, the existence of which can be considered a model example of the routinization of charisma, as expressed by sociologists Weber and Shils. The ʿulamá' converts had a charisma that came not from genealogy (as in the case of the *aghṣán*) or unique gifts (as in the case of the Prophet), but from painstaking study and language skills, as well as an understanding of the ethical and theological implications of sacred texts. While the Shiʿa ulemas contrasted the charisma of office with the Bahá'u'lláh's prophetic charisma, the Bahá'í ulemas took a subservient attitude towards him, contributing their knacks to his disposal.

The final stage of charisma's routinization in the Bahaism was the UHJ. By the will of Bahá'u'lláh, the UHJ has no authority to amend, abrogate, or limit the *'ibádát* (Arabic: عِبَادَات)—the ritual and ethical injunctions relevant to all members of the Faith. Moreover, this elected collegial legislative body could not interpret the remaining obscure passages of the scriptures, as this prerogative was to be fulfilled only by Bahá'u'lláh's sons. The UHJ was planned to be merely a regulatory establishment that issued ordinances relating to new situations or practices that emerged later and were not clearly explained in the scriptures' texts (Cole 2005, p. 321). Thus, the authority of the UHJ bears rational–legal features. It is derived from the office charisma, not from the individual charisma of the founder–Prophet.

In short, the UHJ constitutes law but does not construct creed. However, it is impossible not to notice that in the Bahá'í administrative hierarchy, the UHJ works as a bridge between the human soul and the Covenant of Bahá'u'lláh, thus having the hallmarks of a mystical adhesive (Sadeghzade Milani 2002, pp. 98–99). Although there are several individual levels of authority between the founder and the collegial official body of the Faith, the spiritual Covenant, administered by the UHJ, serves to maintain the accordance of the global Bahá'í community. For the Bahá'ís, the UHJ, as well as their Local or National Spiritual Assemblies, are not bureaucratic administrative bodies commonly elected in a three-stage process, but rather a sacred repository of divine guidance and the spiritual axis of the world.

Bahá'u'lláh, through the announced collegiate authority of the UHJ (he wrote: Blessed are the rulers and learned among the people of Bahá), expressed a vision of religious law in which the operation and application of religious law must affirm, strengthen, and cultivate the central human capacities for love and knowledge (Bahá'u'lláh 2005, pp. 215–16). Despite charisma's obvious routinization, it cannot be stated that authority is based just on legalistic, rule-oriented, and fear-based premises (Danesh 2014, p. 44). Currently, in the Bahá'í Faith, the application of regulations is separated from the authority of someone in charge. Moreover, the political segment of Bahá's teachings have remained unsystematized, rather sketchy, and undeveloped (Грачева and Мартыненко 2015, p. 69).

The order constructed in this way contained crevices. The inadequacy of the seven-part scheme of the authority's hierarchy was already exemplified in 1892, after the Bahá'u'lláh's death, who, while he lived, was the binder and authoritative reference of appeal in all current altercations (Cole 2005, pp. 331–34; Nicolas 1933, pp. 171, 173; Smith 2000, p. 115). The proliferation of sources of authority, according to Shoghí Effendí, was intended to prevent the despotism of the exclusive heir of charisma. Assemblies consist of several people who do not have particular authority; Bahá'ís vote with a pure heart and conscience (Пивоваров 2014, p. 58). Decisions are made in group consultations, during which, unanimity is sought with the approval of a majority vote (Schweitz 1994, p. 44). The cementing adhesive is a common goal and strategy, and the Prophets' charisma continues to be a worldview source.

Ordinances and institutions, from Bahá'u'lláh's standpoint, can only become truly efficacious when the inner spiritual life is purified and transformed. Otherwise, religion will degenerate into a wishy-washy, ossified organization and become something dead (Sadeghzade Milani 2002, p. 101). Nevertheless, it remains indefinite how believers should behave in the occurrence of a possible disagreement between different types of charismatic authorities, as happened between Bahá'u'lláh's sons just after his death (Scharbrodt 2008, pp. 89–93; Wilson [1915] 1970, pp. 252–59). This should not come as a surprise: the succession predicament related to the change in religious leadership was inherited by Bahaism from Shiʿism, which was the matrix of this denomination (Adamson 2007, pp. 116–18; The Universal House of Justice 1992; Sects of Bahá'ís n.d.).

The Bahá'í Faith is founded on the premise of the divinity of the Divine Messenger, so the source of the authority of the *Kitáb-i-Aqdas* for Bahá'ís is the charisma of this text's author. Anyone who acknowledges Bahá'u'lláh as the Lord of the Age will "observe commandments, for the love of [His] beauty", even if this will "cause the heaven of every religion to be cleft asunder" (Bahá'u'lláh 2005, pp. 2–3). It is easy to understand that recognizing and obeying the decrees of Bahá'u'lláh are inseparable dual commitments that the followers of the Faith are obliged to fulfill. And vice versa: for those who do not admit the authority of the Prophet Bahá, these injunctions are absolutely irrelevant.

Bahá's concept of unity is reminiscent of the ideational, transcendental harmony of religion inherent in the divine plan of salvation. The Covenant concept is a relationship between God and humans, and the commitments of humanity in fulfilling the elements of this Covenant emphasize the accordance that exists among the Divine Messengers. To be precise, personal salvation is not intended only for those who believe in Bahá'u'lláh (Brogan 2003, p. 13; McLean 2008, p. 263; Лалуев 2016, p. 233). The Faith leader directly encouraged people to open their ears and listen to the word of God (cf. Rev 3:22), recognizing his teaching as the word of God (Bahá'u'lláh 2011, pp. 36, 81, 98).

Against this background, Bahá'u'lláh's revelation should be seen as a strong and radical break with the then-dominant Shiʿite statement of religious authority. It is true that compared to his rival brother, the radicalism of the Bahá'í Faith's founder was not so evident (Иоаннесян 2003, p. 72). The reticent but contrarian Ṣubḥ-i Azal represented the conservative core of the original Bábí movement, remaining opposed to novelty and the preaching of religion to a non-clergy elite. Ṣubḥ-i Azal's more traditionalist line was unappealing to most Bábís, who were disillusioned with the fateful consequences of the early uprisings and the scale of persecution.

Otherwise, Bahá'u'lláh, ambitious, but at the same time peaceful, exemplified a return to the earlier Shiʿite ideals of political quietism and peaceful standards of propagating the new creed (Wilson [1915] 1970, pp. 135–37). In their ideas for rebuilding the world, Bahá'ís focused on spiritual internal transformations. Bahá'u'lláh wrote: Possess a pure, kindly and radiant heart, that thine may be a sovereignty ancient, imperishable and everlasting (cf. Ps 51(50):12a; Bahá'u'lláh 2011, p. 10). At that time, extremist Azali Bábí were antagonistic towards their surroundings: many of the activists and agitators of the Iranian Constitutional Revolution of 1906–1911 were Azalites (Dabashi 2011, pp. 203, 236; Yazdani 2005, pp. 174–88).

It is easy to see that history has come full circle: the Bahá'ís, in declaring neutrality—set not your affections on mortal sovereignty and rejoice not therein—essentially returned to the ideas of Shaykh al-Aḥsá'í. Shaykh al-Aḥsá'í called for both moral renewal and spiritual growth, not political involvement (Abassy 2010, pp. 90–91; Bahá'u'lláh 2011, p. 114; MacEoin 1983, pp. 222–23). Instead of Weber's omnipotence of a leader, we are dealing here with charm and moral perfection, as described by Freud.

Bahá'ís should follow local laws and obey civil authorities. However, it is permissible to try to convince people who decide on the shape of the law through explications and explanations, but without pushing personal arguments under any circumstances. If the constituted footnotes violate the standards of the Faith in a blatant manner, one should emigrate, but not take up an armed struggle. Disloyalty to the rightful sovereign is

disloyalty to God himself and the love of one's motherland is part of faith in God (cf. Rom 13:1–7; Żuk-Łapińska 1993, pp. 37–38). Further, Bahá'u'lláh erased the category of holy war from his writings (Dehghani 2014, pp. 25–26; Wójtowicz 2021, p. 104).

In the eyes of the Bábís, Ṣubḥ-i Azal appeared to be just the formal head of the community, an exoteric figurehead, while Bahá was recognized as the authentic, esoteric leader (Cole 2004, p. 229). Likewise, Azal's pullback, along with his reluctance to continue fighting, lessened his popularity among the Bábís. Beginning in 1863—the time of his declaration of being the bearer of a new revelation—Ḥusajn-'Alí began to receive pleas from converts for advice.

After a decade, the time had come: the Most Holy Book denied the truth of orthodox Islamic concepts, and the author's attitude towards the revelation of new laws began to take on the features of a radical break (cf. Surah 13:39). And if the Báb's proclamations and claims primarily attacked Islam, Bahá'u'lláh's message was built on attacking the role of clerical authorities in other faith systems as well. The Bahá'ís' leader showed his brother Azal as an example of a narrow and selfish religious hierarchy, presenting him as someone proud and haughty before God. Such a dual theme of the struggle for dominance and doctrinal orientation would be repeated later in the contest between 'Abdu'l-Bahá and his younger half-brother, Muḥammad ʿAli (Berger 1957, pp. 101–2).

Undoubtedly, the prior tenet and lawful axiom of the Bahá people's head has become the elimination of all spiritual authorities. Like the Báb who moderated his demands, Bahá'u'lláh also applied the gradualism principle (Danesh 2014, p. 33). This is nothing other than the phenomenology of being oneself when the leader assigns actions following immediate axiological norms, dosing them depending on the level of preparation of the recipients of the charismatic message content.

The linguistic issue cannot be ignored: the text of the *Kitáb-i-Aqdas* is full of synonyms for the noun 'law': decree, order, commandment, and direction. According to the author, this was to overthrow the legalistic orientation of the Shiʿite tradition. 'Abdu'l-Bahá wrote after his father's death that proper spiritual leaders have no *relationship* to corporeal concerns, *affairs* of *political leadership*, or *worldly matters* (Cole 2005, p. 323).

Unlike his predecessor, Bahá'u'lláh claimed the fullness of his revelation. He wrote: whoso layeth claim to a Revelation direct from God, ere the expiration of a full thousand years, such a man is assuredly a lying impostor (cf. Gal 1:7–8; Bahá'u'lláh 2005, p. 11). On the one hand, such a claim put an end to dissensions about the possible arrival of a prophet after the Bahá'u'lláh's death. On the other hand, it allows us to conclude that Bahaism was not a branch of Babism, but a distinct religion (Effendi 1991, pp. 107–12; McLean 2008, p. 246; Мартыненко 2018, p. 238).

Researchers compare the Báb to John the Baptist, whose community of disciples almost ceased to exist with the proclaimed Revelation of Jesus. If Christ is considered wine, then Bahá'u'lláh is considered the lord of the vineyard among his devotees (Wilson [1915] 1970, pp. 35, 41, 95–98). Hereby, by referencing figures from other religions, Bahá'ís encourage and inspire their adherents, placing recent actions of the Faith against the background of noble tradition, shaping the collective memory of the history in their believers' community (Hassall 2008, p. 76). This assumes that the multiplicity of world religions, according to Bahá'u'lláh, agrees with the divine plan for the salvation of humanity, encompassing all religious systems, which illustrates the humanocentrism of the Faith (Pope Francis 2021, p. 398; Wójtowicz 2021, p. 100).

## 3. Materials and Methods

The issue of charisma, which was analyzed in this paper, was not chosen randomly. Personal charisma falls into categories to which the methods of scientific analysis can be applied. Certainly, it was essential to show charisma's background, highlight various forms of its manifestations, determine the causes of the disappearance of charisma, and, finally, determine the practical significance of studies on the charisma phenomenon in the aspect of religious reality transformation.

The methodological approach was based on understanding the internal structure of the matter, highlighting the social system hierarchy within which the charismatic personality is integrated, with an emphasis on the instability of this system and the ability to transform, including due to the depersonalization of charisma. General logical methods were used (abstraction and generalization, induction and deduction, analogy, analysis, and synthesis), as well as historical and cultural analysis methods.

Each religious association is born in characteristic cultural conditions and a specific historical period. By analyzing Babism/Bahaism source materials about the Faith founders, which are hagiographic in nature, we are able to understand one of the main goals of this genealogy. Portrayals of the leader's life take the reader on a spiritual pilgrimage, which is a metaphor for the evolution of life's search. These texts allow us to comprehend how the Faith originators grappled with others or with adversities, overcame passions, overcame crises, and strove to popularize their teaching (Hassall 2008, p. 82). The presentation of the key founding figures takes place in two modes: heads are presented as authentic people acting in specific circumstances and, at the same time, as an imaginary image whose personification was expected. In this way, the charismatic figure with divine attributes becomes someone with easily recognizable and approachable markers (Rothstein 2016, p. 398).

Through the method of comparative analysis, it is possible to notice the similarity between the key figures of the Faith. A review of Bahá'u'lláh's messianic and prophetic claims allows us, in a broader context, to apprehend the offer of leaders of new religious movements who articulate their eagerness to challenge existing standards or institutions, while offering solutions to troubling current questions. It was easy for the teachings' adherents of the new revelation to accept Bahá'u'lláh's claims because his authority as a charismatic leader somehow overlapped with the prerogative of abrogating Islamic laws, which the Báb, his predecessor, had already used.

Both leaders' deaths, although they triggered the expected succession clashes, did not cause the founder's charisma to fade. Bahá'ís like to emphasize that the primary mission of the Báb's life was to know and love God. Not even an arrest or the subsequent death penalty prevented him from preparing the path for those who came after him. The Báb, the holy Messenger of God, is a leader for whom martyrdom became throwing oneself at the feet of the Beloved God with all felicity (Esslemont 1992, p. 31).

## 4. Discussion and Results

The figure of Bahá'u'lláh in Bahá'í doctrine has a dual dimension: theandric or divine-human. As a man, he carried out the mission entrusted to him to convey news about the Most High to other people, and another part of his charisma was his prophetic statements when he conveyed word from the Most High. In the first status, he embodies a leader in a state of deepest humility, while in the second, his human personality becomes totally subordinate because, as the Bahá'ís believe, God communicates through him. It seems intriguing to explore more deeply the texts of Bahá'u'lláh, in which the human narrative is often mixed with the revelation of God's truths as if the Most High were speaking in the first person (MacEoin 2009, pp. 50–54). Moreover, in Bahá'u'lláh's example, we can see the classic model of relinquishing administrative roles while retaining charismatic authority (Bromley 2016, p. 115).

Nonetheless, the fundamental dissimilarity between the charisma of the Báb and Bahá'u'lláh is that the authority of the Prophet of Babism was part of Muslim messianism, where the guiding idea was waiting for the *Mahdí*; in this matter, it was necessary to resist and proclaim the Báb's preaching. Bahá'u'lláh acted differently, as he was closer to the idea of giving his life for the faith than to taking the same life of another. Ḥusajn-'Alí did not believe in the continued presence of the twelfth imam and his quite long life, and he considered the tradition regarding the figure of *al-Qá'im* to be devoid of truth (Eschraghi 2014, pp. 123, 125; Грачева and Мартыненко 2015, p. 76).



The *Mahdí* in the Bahá'í Faith ceased to be the highest authority, the axis of the universe around which all debates centered, and remained rather a relic from ancient times. The use of this title was merely pragmatic or didactic, but primarily, it was a temporary step to prepare mankind for the true statement. A paradigm shift is evident: if one did not accept and understand Bahá's charisma, one had to pray for him; after all, there would be no other charismatic revelator for the next millennium (Hutter 2009, p. 36; Kokot-Góra 2018, p. 50; Лалуев 2016, p. 239).

The ideological trajectory of Bahá'u'lláh's reformist concept was developed and systematized through the later interpretive writings of 'Abdu'l-Bahá, as well as Shoghí Effendí. Today, it is is being purified through the UHJ decrees and directives. To sum up, it remains to be said that, despite the impression of the melting of Bahá'u'lláh's authority in the matrix of the institutional hierarchy of the Bahá'í Faith, the dynamics of this figure are undoubtedly an interesting example of the charisma of the new religious movement's leader; it is inspiring and worth analyzing by the new religiosity scholars (Bahá'í International Community 2017).

As noted by Christopher Buck, faculty instructor at the Department of Bahá'í History and Texts at the Wilmette Institute in the USA, the development of Ḥusajn-'Alí's charisma passed through consecutive stages: mystic messiah, prophetic messiah, and royal messiah (Buck 2004, p. 143).

The period of mystical charisma corresponds to the Baghdad period (1856–1863), when the messiah's role emerged, although in a hidden way. Bahá'u'lláh was authenticated as a prophet when he promulgated himself as *maẓhar-i Illáhí* (1866–1867), i.e., the ninth tier in the prophetic chain. Finally, in his letters to monarchs and rulers (1866–1873), Bahá identified himself as a globe reformer with a concept for bettering the planet as well as teaching unity, justice, and equality (Bahá'u'lláh 2011, pp. 12, 29, 51, 55; Saiedi 2000, p. 7). In this way, the Bahá'í Faith can be viewed as an anticipation of the processes of religious convergence that would manifest themselves most fully in the second half of the 20th century (Мартинюк and Никитченко 2009, p. 127).

This three-stage division can also be logically derived through a retrospective analysis of the Prophet–founder's mission sequence stages. Originally, Ḥusajn-'Alí addressed the message to his most intimate and loyal group's members—the mystics. Later, he extended the recipients' circle, including the ulemas—the clergy. And finally, he addressed the sovereigns—the world leaders and monarchs. Thus, it can be supposed that the absolute status of Bahá'u'lláh's leadership among the Bábís arose years earlier than it was officially proclaimed (Eschraghi 2014, pp. 112–13). This reasoning also fits into the scheme of sociologists R. Stark and W. S. Bainbridge: first, Bahá'u'lláh became the founder of a new idea (to be more precise, the originator of a new reading of an already existing idea, previously revealed by the Báb), later becoming its transmitter, and finally becoming its apostle.

## 5. Conclusions

It should be stated that after the founder of the Faith's death, the young cult moved from the charismatic leadership period to the organizational phase. Even Shoghí Effendí admitted that the Spirit, animating the world, became incarnate in institutions, confirming the continuity of the mystical connection with Bahá'u'lláh and fulfilling the Master's enactments (Effendi 1991, pp. 4, 19–22). According to the American sociologist Peter L. Berger, the Faith developed from a sect, when charisma was associated with the leader, to the status of the Church, so charisma passed to the office (Fozdar 2015, p. 281). The authority routinization, nonetheless, should not be with the abandonment of intention or spirit by the greatest charismatic characters of the Bahá'í Faith. The process of institutionalization is intended, outlined by Bahá'u'lláh, and therefore essentially imbued with spiritual principles (Лалуев 2016, pp. 230, 284; Melikova 2013, p. 127; Пивоваров 2014, pp. 85–86, 131).

The Bahá'ís believe that any attempt to split the administrative order from the doctrinal teaching should be considered a mutilation of the Cause of God's body. Such manners may

result not only in the disintegration of the parts but even in the extinction of the Faith itself. Nonetheless, the bureaucratic charisma of the UHJ visibly differs from that of the Báb and Bahá'u'lláh, so a closer analysis of the essence of the authority of the Faith's governing body would be beyond the scope of this text.

The Bahá'í Faith is almost always referred to in the literature as the youngest world religion (Hartz 2009, p. 8; Ferrer 2021, p. 253). This new community of believers is defined as *a unique religious movement responding* to globalization processes by creating a worldwide religious identity for its adherents through both ideological and organizational means (McMullen 2000, p. 11).

The founder–Prophet of the Bahá'í Faith played an indispensable role in this, for by proclaiming himself as *Man yuẓhiruhú Alláh*, he announced the fulfillment of the messianic signs of other world religions (cf. Isaiah 9:6; Jn 14:17; McLean 2008, p. 261). In 1925, the Mufti of Egypt recognized the Baha'i Faith as a completely separate religion with other principles, beliefs, and laws different from those of Islam. However, in Iran, this religion is still censored, especially after 1979, because it is perceived as a *political movement created* by the *colonial* governments for the weakening *of Islam*, a false sect and a tool in the hands of foreign powers, and a tool for sowing discord (cf. Jn 1:11; Hosseini 2023, p. 110; Dehghani 2014, pp. 16–17; Melikova 2013, p. 128; Smith 2019, pp. 57–59).

**Funding:** This research received no external funding.

**Data Availability Statement:** Publicly available literature was analyzed in this study. All data used in the article are documented by published sources. No persons other than the authors of the works cited are mentioned in the article. Data sharing is not applicable to this article.

**Conflicts of Interest:** The author declares no conflict of interest.

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
