# Peer review of "Transition, Emulation and Dispute over Authority in the Bábí/Bahá’í Faith"

_religions, doi:10.3390/rel15050577_

Round 1

Reviewer 1 Report

Comments and Suggestions for Authors

This is an important and potentially excellent article. I have two issues with its current condition. The smaller one is that there is a fair number of syntax errors that need fixing--a bit of copy-editing would be useful. More significant is this: the article is intended to be read by someone who is not already well-versed in the history and tenets of the Baha'i faith--as evidenced, for instance, in the laying out of the seven sources of authority. Given this and given that the stated intention is to indicate how Baha'i emerged and evolved in the wake of the death of the Bab, it requires two things that are missing. One is greater clarity with regard to who is who in the initial decades after 1850: contrary to the abstract, it is not simply a clearcut matter of Subh-i 'Azal versus Husayn 'Ali. Two is that key terms, such as Bayan, need to be defined: if I don't know enough about Baha'i, that term is meaningless to me... Similarly, other key words, such as al-Qa'im, should be defined. none of this constitutes major fix.

Comments on the Quality of English Language

See the previous comment: someone needs to take a carefu look at the syntax throughout the article. Not a huge number of miscues, but enough that require fixing!

Author Response

Dear Colleague,

Thank you for your suggestions and remarks. I augmented the introductory part of the paper with data about the social and religious background of 19th-century Persia. Besides, I have added an explanation of terms that may be unclear to those unfamiliar with the Bahá'í Faith.

Best regards!

Reviewer 2 Report

Comments and Suggestions for Authors

The paper provides a reader with an interesting and in-depth analysis of the problem of succession and authority within the tiny community of the followers of the Bab’s teachings. Tracing some historical accounts, it attempts to distinguish and clarify the forms of action undertaken by Mirzā Yahyā Nuri and Mirzā Hoseyn-Ali Nuri respectively. In so doing, the Author uses mostly the published studies what runs the risk of being accused of reducing the importance of the primary sources or writings of  Mirzā Yahyā Nuri and those who supported him. In a sense, the article presents rather the Bahai interpretation of the events. 

The strongest points of the paper include: 

1) The satisfactory argumentation and presentation of the main line of the dispute over the leadership and authority after the Bab’s death.

2) The chosen literature for providing analysis. 

3) The logic of presented material and the good structure of the text.

Still, however, there is room for some improvement.

I suggest expanding the first part of the text by giving more detailed information about the events in the mid-19th century Persia as well as some concepts (or rather variety of concepts) of authority within the Shia Islam as that became the point of reference for the Author’s research. Some methodological inclinations should be added.

I suggest using the standard transcription of the Persian/ Arabic names/ words, for instance the transcription provided by the Library of Congress or Encyclopaedia Iranica can be used. It is necessary to read and check again the Persian/ Arabic words used in the article. The most popular names can be used in their common and accepted forms like Qajar, Naser al-Din Shah, etc.  

Author Response

Dear Colleague,

Thank you for your suggestions and remarks. I augmented the introductory part of the paper with data about the social and religious background of 19th-century Persia. Besides, I added a comparison table with the keywords that are most important in this article. In one column there are words in the transcription that I used in the text, and the next column, I placed the transcription according to Encyclopaedia Iranica.

Best regards!